# Detecting High-Resolution Adversarial Images with Few-Shot Deep Learning

Junjie Zhao [1], Junfeng Wu [2], James Msughter Adeke [1], Sen Qiao [1] and Jinwei Wang [2,3,*]

1 School of Electronics and Information Engineering, Nanjing University of Information Science and Technology, Nanjing 210044, China; ginogogo@nuist.edu.cn (J.Z.); adekejames@nuist.edu.cn (J.M.A.); sensariel@nuist.edu.cn (S.Q.)
2 School of Computer Science, Nanjing University of Information Science and Technology, Nanjing 210044, China; wjf@nuist.edu.cn
3 State Key Laboratory of Mathematical Engineering and Advanced Computing, Zhengzhou 450003, China
* Correspondence: wjwei@nuist.edu.cn; Tel.: +86-138-5199-4653

**Abstract:** Deep learning models have enabled significant performance improvements to remote sensing image processing. Usually, a large number of training samples is required for detection models. In this study, a dynamic simulation training strategy is designed to generate samples in real time during training. The few adversarial examples are not only directly involved in the training but are also used to fit the distribution model of adversarial noise, helping the real-time generated samples to be similar to adversarial examples. The noise of the training samples is randomly generated according to the distribution model, and the random variation of training inputs reduces the overfitting phenomenon. To enhance the detectability of adversarial noise, the input model is no longer a normalized image but a JPEG error image. Experiments show that with the proposed dynamic simulation training strategy, common classification models such as ResNet and DenseNet can effectively detect adversarial examples.

**Keywords:** deep learning; anomaly detection; adversarial example; high-resolution image; image processing

## 1. Introduction

With the popularity of artificial intelligence technology, deep learning models have enabled revolutionary performance improvements in the fields of remote sensing image processing, such as semantic segmentation [1–3], classification [4–7], and target detection [8,9]. However, adversarial examples [10] pose a significant threat to the security of artificial intelligence models such as convolutional neural networks (CNNs). By simply adding a small adversarial noise to a normal picture [11,12], CNNs can be guided to an incorrect conclusion. Training an effective adversarial example detection model typically requires many labeled samples [13,14]. Obtaining a usable detection model with only a small number of samples is necessary.

The defense methods for adversarial examples are mainly divided into active and passive defense. Detection is a typical active defense method. The addition of adversarial noise destroys the original distribution of the image, making statistical values of adversarial examples different from the raw image in the spatial and transformation domains. Grosse et al. [15] found that the maximum, average, and energy distribution differences of pictures can be used to detect adversarial examples. Li and Li [16] identified adversarial examples using the internal characteristics of the attack model and support vector machine (SVM). Their method directly obtains the output feature maps of each convolution layer of the original model and sends the statistical features of these feature maps to the SVM for classification. However, their performance needs to be improved. Liu et al. [17] proposed an enhanced version of the spatial rich model (ESRM) in combination with steganalysis,

which achieved high-accuracy detection. Zhao et al. [14] used a simulated JPEG quantization error for detection, and their findings revealed that the block discrete cosine transform (DCT) domain is more suitable for detecting adversarial noise than the spatial domain. Effective detection models must use a large number of adversarial examples for training.

With the development of adversarial example generation technology, researchers have generally sought to reduce the magnitude of adversarial noise to reduce the changes in visual effects. However, adversarial noise is reduced at the expense of increasing the number of generation steps. The fast gradient sign method (FGSM) [10] only needs to add noise once to complete the attack. Subsequently, the number of iterations for the other generation methods increases exponentially [18,19]. This undoubtedly introduces significant challenges to the generation of large-scale training samples.

Adversarial training is an effective passive method to defend against adversarial attacks [20,21]. In the adversarial training process, training samples must be generated in real time, and the direct use of the original generation method slows the training process. Therefore, training sample generation and defense in adversarial training are usually asynchronous [11,22]. Inspired by this, we count the distribution characteristics of adversarial noise with a few examples and use many clean pictures to generate pseudoadversarial examples. Thus, we can provide a large number of training samples for the detector. The proposed dynamic simulation training strategy for adversarial example detection models is characterized by the following innovations and advantages:

- Pseudoadversarial examples are generated through simulation to train the detector so that the training model no longer requires many real adversarial examples. A small amount of real adversarial noise is used to fit the distribution, and pseudoadversarial examples are generated based on this distribution;
- A dynamic simulation training strategy for the entire detector training process is proposed based on pseudoadversarial examples. With the cooperation of a small number of real examples, JPEG error input, and compression factor fluctuation, the dynamic simulation training strategy helps common classification models to detect adversarial examples;
- Because pseudoadversarial examples are generated in real time during the training process of the detection model and the noise generated in each epoch is different, the serious overfitting phenomenon is avoided. Benefiting from this, the common classification model performs well in adversarial example detection.

The remainder of this paper is structured as follows. In Section 2, the generation of adversarial examples, defense, and other related works are introduced. Section 3 introduces the relevant basis of JPEG compression error, and in Section 4, we introduce the proposed training scheme. In Section 5, detailed experiments and analyses are presented to verify the feasibility of the proposed scheme. Finally, we conclude this paper in Section 6.

## 2. Related Works

### 2.1. Adversarial Example Generation

An adversarial example is the addition of artificially designed small adversarial noise to normal examples, such as clean pictures, to mislead machine learning models. Let $X_c$ denote the normal input sample and $r$ be the adversarial noise. The adversarial example ($X_{ad}$) can be obtained as shown in Equation (1). The amplitude of adversarial noise is usually so small that it is imperceptible to human vision.

$$X_{ad} = X_c + r \tag{1}$$

The generation methods for adversarial examples are mainly divided into white-box and black-box attacks. In white-box attacks, the parameters of the target model are known to the attacker. In contrast, a black-box attacker knows only the output of the target model (discriminative results or confidence levels).

While confirming the existence of adversarial examples, Goodfellow et al. proposed FGSM to rapidly generate adversarial examples [10]. They directly superimposed the gradient of the model on the input image, causing the diffusion of the loss function and misleading the model. Equation (2) describes the generation process, where $y$ represents the output label of the input ($X_c$) of the target model, $\nabla$ represents the gradient calculated to $X_c$, $L$ represents the current loss function, $sign$ indicates the sign function, and $\epsilon$ represents the perturbation coefficient.

$$X_{ad} = X_c + \epsilon\, sign(\nabla_x L(X_c, y)) \tag{2}$$

The FGSM only needs to generate and add noise once to generate adversarial examples, which is a rapid process. However, its perturbation coefficient ($\epsilon$) must be set artificially, and its attack success rate on large-scale images is low. The schemes that add noise in multiple steps effectively solve these problems. The basic iteration method (BIM) [23] adds gradient noise in several steps and recalculates the gradient direction after each step. Let $X_N$ denote the sample generated at the $N$ step, $\alpha$ be the single-step noise amplitude, and $Clip_\epsilon$ denote the truncation of cumulatively added noise by $\epsilon$. The samples generated by the $N + 1$ step of BIM can be represented by Equation (3), where $X_0 = X_c$.

$$X_{N+1} = Clip_\epsilon\{X_N + \alpha\, sign(\nabla_x L(X_N, y))\} \tag{3}$$

Dong et al. proposed an optimized version of the FSGM based on momentum iteration (MI-FGSM) [24]. Since then, many multistep generation methods have been designed to achieve two effects: a higher attack success rate and a lower noise amplitude. Compared with BIM, project gradient descent (PGD) [11] uses less single-step noise and more iteration steps. To avoid long generation time, the first step of the PGD attack is to add random noise to the input sample. Currently, PGD remains one of the most effective attack methods. The C&W method [12] can effectively improve and reduce the amplitude of adversarial noise by limiting the norm of adversarial noise ($L_1$, $L_2$, and $L_\infty$) while ensuring the attack effect. The Deepfool method [18] assumes that the classification space of the target model is a linearly differentiable space. In each iteration, the algorithm gradually moves the input image to the decision boundary until the image is finally moved to the other side of the decision boundary.

Compared to white-box attacks, black-box attacks can only use limited information. The strategies for black-box attacks include gradient estimation and boundary queries. Brendel et al. [25] transformed the estimation problem into an optimization problem and realized gradient estimation using prior knowledge such as time correlation and data correlation. The boundary query method, also known as a decision-based attack, was first proposed by Brendel et al. [26]. A boundary attack [26] needs to randomly generate an adversarial example, then iteratively reduce the noise to be closer to the original sample.

In addition, there is a series of generation methods based on generative adversarial networks (GANs) [27]. Xiao et al. [28] mapped original samples into adversarial disturbances using the GAN generator and added them to the original samples. The discriminator determines whether the input samples are adversarial examples. Mangla et al. [29] introduced an inner convolution layer in a classifier to extract features. They input the features and random noise into the generator of GAN and proposed AdvGAN++. During deployment, AdvGAN [28] can perform black-box attacks. However, AdvGAN++ [29] can only carry out white-box attacks because it requires the output of the middle layers of the target model.

### 2.2. Defense of Adversarial Examples

Defense methods for adversarial attacks include detection, repair, input transformation, adversarial training, etc. Feinman et al. [30] proposed measuring the distances between adversarial and natural examples using kernel density estimation of the classifier's hidden layers. Carrara et al. [30] proposed a method to extract the output of the classifier's hidden layer neurons, then used a long short-term memory (LSTM) network

to detect adversarial examples. Schottle et al. [31] used a steganalysis tools to detect PGD adversarial examples. SmsNet [13] is an end-to-end adversarial example detection model, and the feature statistical layer was designed to obtain the high-dimensional features of each convolutional layer's output.

The repair of adversarial examples mainly uses downsampling and reconstruction. In ComDefend [32], ComCNN was first used to compress the adversarial example from 24 bits to 12 bits; then, ResCNN was used to restore the compressed image to a high quality. SmsGAN [33] downsampled examples in the spatial dimension and reconstructed high-resolution images using GAN. Operations such as flipping, scaling, and filtering can mitigate the effects of adversarial attacks [34]. The experiments of Kurakin et al. [23] showed that JPEG compression decompression significantly resists adversarial attacks.

The goal of adversarial training is to obtain a robust model that is difficult to successfully attack. Therefore, this defense approach assumes that the attacker and the defender use different adversarial attack methods. While proposing PGD attacks, Madry et al. [11] suggested using samples generated by the PGD for adversarial training. By constraining the logit distance between the output of the natural example and its corresponding adversarial example, the logit pairing method [35] improves the robustness of the model and reduces its influence in recognizing normal inputs. Zhang et al. found that too many iterations of generating training samples affect the model's convergence. They proposed friendly adversarial training (FAT) [21] using a smaller number of iterations and gradually increasing this number while generating training samples with PGD.

### 3. Prior Knowledge

#### 3.1. JPEG Encoding and Decoding

JPEG compression is performed in the DCT domain and the YCbCr color space. As shown in Figure 1, the JPEG compression process includes block, color space conversion, DCT, quantization, and entropy encoding [36]. Entropy encoding includes zigzag and Huffman coding methods.

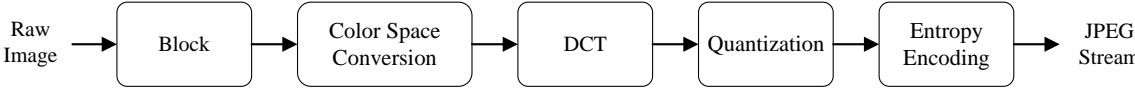

**Figure 1.** JPEG compression process.

Corresponding to the compression process, the JPEG decompression process includes entropy decoding, inverse quantization, inverse DCT (IDCT), color space inversion, and block combination, as shown in Figure 2.

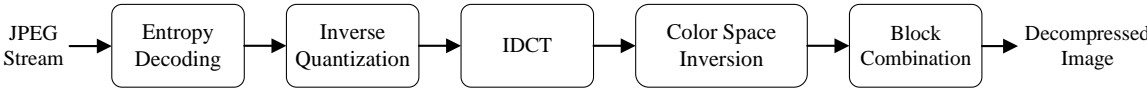

**Figure 2.** JPEG decompression process.

During the compression process, block, DCT, and entropy coding are lossless. However, the quantization process incurs significant loss, and the color-space-converted data must be converted from float numbers to eight-bit integers. Therefore, JPEG compression is lossy. Different quantization tables are used according to different quality factors, resulting to different loss magnitudes.

#### 3.2. JPEG Error and Adversarial Example Detection

In the JPEG compression and decompression process, quantization, conversion, truncation, and rounding errors [37] are generated. These errors are collectively referred to as JPEG errors. Using JPEG compression as an input transformation can effectively resist adversarial examples, since adversarial noise is lost in the JPEG compression and decom-

pression processes. Therefore, inputting JPEG errors into the detectors can improve the detection effect of adversarial examples.

We used the FGSM ($\epsilon$ = 2) to generate 30,000 adversarial examples. Using the corresponding 30,000 original images, a dataset of 50,000 training images and 10,000 test images was formed. We used JPEG errors (quality factor = 100) and normalized images to train VGG16 [38] on the dataset. The test results for different epochs are shown in Figure 3.

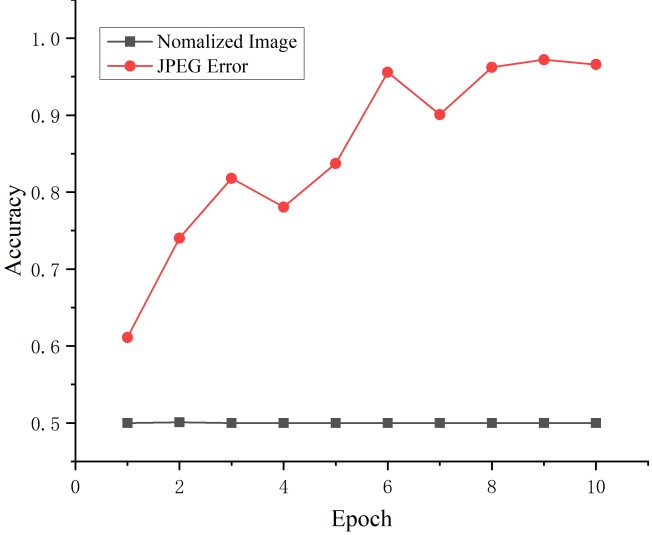

**Figure 3.** Accuracy of VGG16 for JPEG error and normalized image input after each training epoch. With the training, only the accuracy curve with JPEG error as input shows an upward trend.

As shown in Figure 3, the model with input of normalized images is entirely unable to recognize adversarial examples. At the same time, the model with JPEG error can distinguish adversarial examples from natural examples. Zhao et al. [14] simulated the JPEG compression process and found that the YCbCr block DCT (bDCT) domain is more efficient in adversarial example detection than the RGB space domain.

## 4. Dynamic Simulation Training Strategy

### 4.1. Problem Description

Equation (1) shows that an adversarial example can be obtained by adding noise to the natural example. The main method for detecting adversarial examples is to identify adversarial noise. The symbol $\sim$ indicates obedience to a particular distribution. Assuming that the amplitude distribution of adversarial noise conforms to a special probability model ($M$), the purpose of training the adversarial example detection model can be described by Equation (4), where $m \sim M$, *model* represents the detection model, and $D_{KL}$ represents the Kullback–Leibler (KL) divergence.

$$max\ D_{KL}(model(X)||model(X+m)) \qquad (4)$$

Since the probabilistic model ($M$) is unknown, we can only estimate it from a finite number ($m$). The distribution model ($M_2$) satisfies Equation (5) and can be obtained by fitting the statistical data. Then, the new purpose of training shown in Equation (6) is designed, where random noise $m_2 \sim M_2$.

$$\exists\ m \sim M,\ m \sim M_2 \qquad (5)$$

$$max\ D_{KL}(model(X)||model(X+m_2)) \qquad (6)$$

When an accurate $M$ distribution cannot be obtained, the parent distribution ($M_2$) of $M$ can be fitted according to a finite number of $m$. Subsequently, it is only necessary

to follow the $M_2$ distribution to generate data $m_2$ to train the detection model, and the detection of $m$ can be realized.

*4.2. Overall Process*

The dynamic simulation training scheme designed in this study estimates the distribution model by counting the noise distribution of a small number of adversarial examples, then generating pseudoadversarial noise to help train the detection model, as shown in Figure 4. First, the distribution of adversarial noise is counted, and the residual sum of squares (RSS) is used to fit the distribution model ($M_2$). Then, random noise $m_2 \sim M_2$ is generated and superimposed onto clean samples. After obtaining the residual through JPEG compression and decompression, pseudoadversarial examples are input into the classification network for training. To enhance the variation in the input and mitigate overfitting, the quality factor used in the compression–decompression process changes slightly.

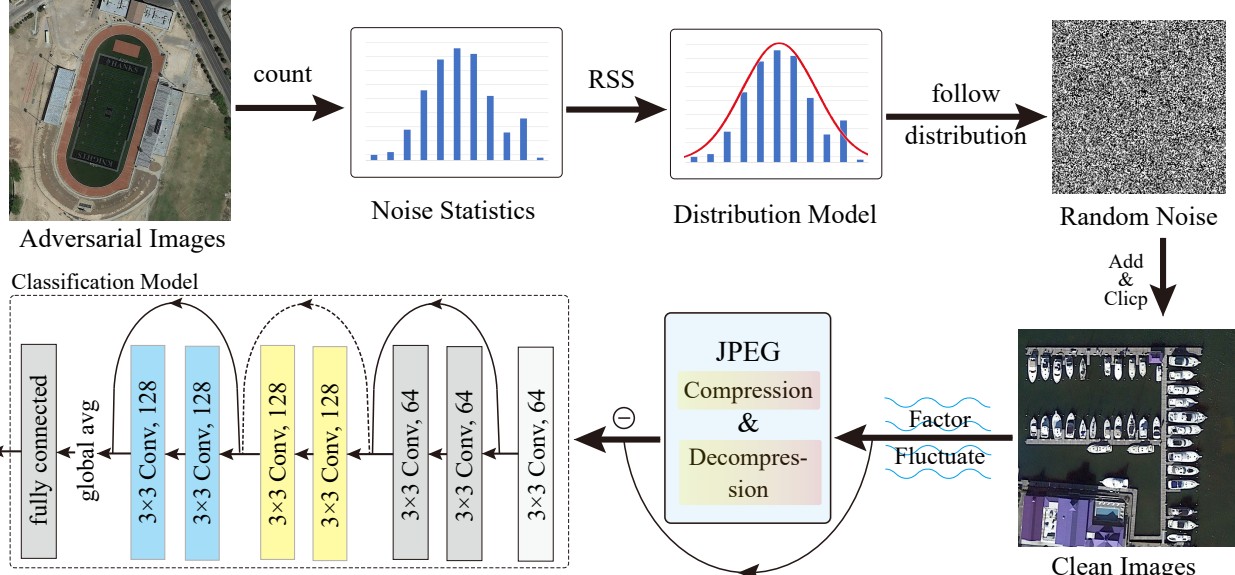

**Figure 4.** Overall process of the dynamic simulation training strategy. $\ominus$ indicates the calculation of residuals between matrices.

*4.3. Characteristic Analysis*

4.3.1. Single Sample Characteristics

A digital image is a two-dimensional discrete signal. The amplitude of adversarial noise, which is stored and transmitted by the image, must be discrete. Because the range of adversarial examples needs to be truncated, the adversarial noise amplitude at the boundary of the pixel value (0 and 255) is irreversibly discarded. Therefore, we exclude pixels at the boundary of the pixel value before analyzing the characteristics. This step is known as statistical cleaning.

The number of adversarial noise values ($N_v$) in the sample is finite and satisfies $0 < N_v < 511$. In each sample, the frequency of each noise value is counted separately.

The distribution of adversarial noise is typically related to the texture complexity of an image. We use horizontal Sobel operator $SobelX$ and vertical Sobel operator $SobelY$ to extract the gradient information of the images and estimate their textural complexity. The calculation process for complexity ($ComX$) is shown in Equation (7), where $\alpha$ is the horizontal and longitudinal complexity coefficient that satisfies $0 < \alpha < 1$.

$$ComX = \sqrt{\alpha||SobelX(X)||_2 + (1-\alpha)||SobelY(X)||_2} \qquad (7)$$

To simplify the calculation, we use the $L_1$ norm to approximate the $L_2$ norm. Then, Equation (8) is used to calculate the approximate complexity ($ComEX$).

$$ComEX = \alpha|SobelX(X)| + (1-\alpha)|SobelY(X)| \tag{8}$$

### 4.3.2. Cross-Sample Characteristics

The frequency of occurrence of each noise value in the adversarial noise of the sample is calculated. For the same noise value with different frequencies, we use the RSS to fit the distribution of each value on multiple samples, as shown in Figure 5. Probability distribution models include normal, exponential, t, beta, gamma, and log gamma distributions.

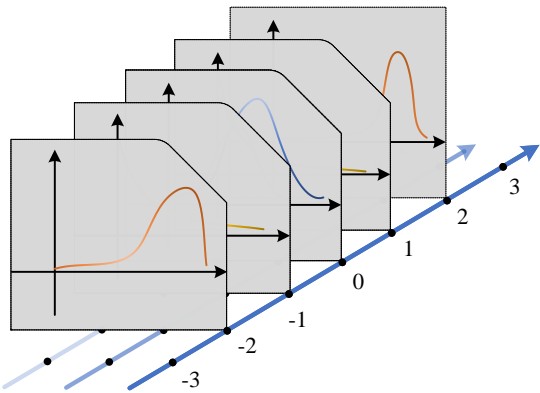

**Figure 5.** Fit of the distribution of each value across samples. Each blue line with an arrow represents the noise in a sample.

BIM is used to generate 1000 adversarial examples, and the distribution of the numbers of value 1 noise is displayed in Figure 6. For the same noise value, the distribution models in the texture and smooth regions exhibit obvious differences.

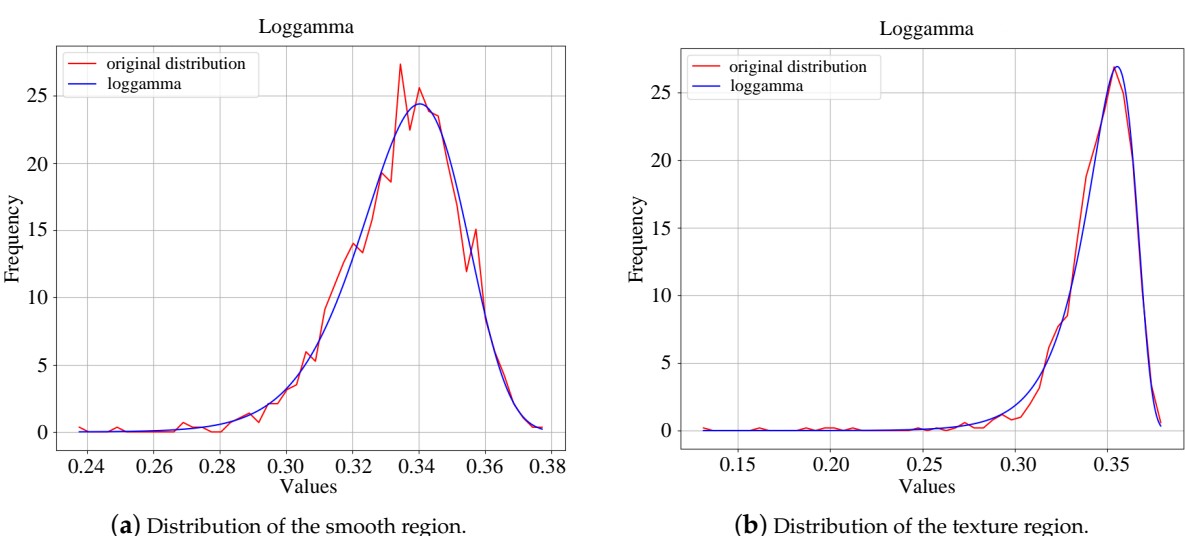

(**a**) Distribution of the smooth region.　　　　(**b**) Distribution of the texture region.

**Figure 6.** Distribution of the numbers of value 1 noise.

To avoid excessive time complexity, we simply use the threshold $T_X$ to distinguish between the texture and smooth regions. The adversarial noise distributions of the texture and smooth regions for each sample ($X$) are counted separately. The results of the characteristics are not very sensitive to the value of $T_X$.

### 4.4. Preprocessing Method

Before training the detection model, the main preprocessing steps include fitting the noise distribution, generating pseudoadversarial noise according to the distribution, and calculating the JPEG error. The steps for fitting the noise distribution and generating

pseudoadversarial noise are described in detail in this subsection. The symbols used in this study are summarized in Table 1.

**Table 1.** Meaning of symbols.

| | |
|---|---|
| $X_{ad}$ | adversarial example(s) |
| $X_{ad}^c$ | clean example corresponding to adversarial example |
| $X_c$ | clean example |
| $Y_c$ | label(s) of $X_c$ |
| $x[\text{i}]$ | the $i$th element in matrix $x$ |
| $f_t$ | JPEGquality factor used when testing |
| $N_t$ | number of training epochs with $X_{ad}$ |
| $*$ | multiplication of corresponding elements in matrices |
| $back()$ | back propagation |
| $cat()$ | combine the input data |
| $clear()$ | statistical cleaning |
| $cross_e(x_1, x_2)$ | cross entropy of $x_1$ and $x_2$ |
| $follow(x)$ | generate random numbers that follow the $x$ distribution |
| $jpeg(x, f)$ | JPEG compression and decompression with quality factor $f$ |
| $len(x)$ | number of elements in matrix $x$ |
| $minibatch(x)$ | select a batch of data from $x$ |
| $model(x)$ | classification model with input $x$ |
| $oneslike(x)$ | all-1 matrix with the same size as $x$ |
| $random.choice(x)$ | randomly select an element from $x$ |
| $range(x)$ | natural number sequence from 0 to $x$ |
| $zeroslike(x)$ | all-0 matrix with the same size as $x$ |

Fitting the distribution model of adversarial noise is the basis for generating pseudoadversarial noise. As shown in Algorithm 1, the fitting process includes adversarial noise extraction, statistical cleaning, texture region division, and RSS fitting.

According to Algorithm 1, the distribution models of the texture and smooth regions are extracted. Furthermore, we analyze the textural complexity of the clean example ($X_c$) and generate each numerical probability of a single example based on the above two distributions. These probabilities are used to generate the final pseudoadversarial noise. The process of generating pseudoadversarial noise for $X_c$ based on the existing distribution is shown in Algorithm 2.

---

**Algorithm 1:** The fitting of adversarial noise distribution

---

**Input:** $X_{ad}$, $X_{ad}^c$

**Output:**

$Dist_{text}$: distribution of texture region

$Dist_{smooth}$: distribution of smooth region

1   $Noise_{ad} = X_{ad}$ - $X_{ad}^c$ ;

2   $Noise_{ad}$ = clear($Noise_{ad}$) ;

3   $X_{text}$ = ComEX($X_{ad}$) ;

4   $Mask_{text}$ = zeroslike($X_{text}$) ;

5   $Mask_{smooth}$ = oneslike($X_{text}$) ;

6   **for** *i in range(len($X_{text}$))* **do**

7     **if** $X_{text}[i] > T_x$ **then**

8       $Mask_{text}[i] = 1$ ;

9       $Mask_{smooth}[i] = 0$ ;

10      end

11    end

12   $Dist_{text}$ = RSS($Noise_{ad}$ * $Mask_{text}$) ;

13   $Dist_{smooth}$ = RSS($Noise_{ad}$ * $Mask_{smooth}$) ;

14   return

---

**Algorithm 2:** Generation of pseudoadversarial noise

---

**Input:** $X_c$, $Dist_{text}$, $Dist_{smooth}$

**Output:**

$Noise_{ad}^p$: pseudoadversarial noise

1   $X_{text}$ = ComEX($X_c$) $Mask_{text}$ = zeroslike($X_{text}$) ;

2   $Mask_{smooth}$ = oneslike($X_{text}$) ;

3   **for** *i in range(len($X_{text}$))* **do**

4     **if** $X_{text}[i] > T_x$ **then**

5       $Mask_{text}[i] = 1$ ;

6       $Mask_{smooth}[i] = 0$ ;

7   $Noise_{text}$ = follow($Dist_{text}$) * $Mask_{text}$ ;

8   $Noise_{smooth}$ = follow($Dist_{smooth}$) * $Mask_{smooth}$ ;

9   $Noise_{ad}^p$ = $Noise_{text}$ + $Noise_{smooth}$ ;

10   return

---

In lines 10 and 11 of Algorithm 2, the follow(.) function includes two steps: generating the probability corresponding to each noise value according to their distribution and generating random numbers according to this probability. Let $P_i$ represent the probability of noise value $i$. In the first step, three different random probability cases are considered. The corresponding processing strategies are as follows:

- $\exists\, P_i < 0$ : repeat step 1;

- $\forall\, P_i \geq 0\ \&\ \Sigma_{i=1}^{N_v} P_i \leq 1 : P_0 = P_0 + (1 - \Sigma_{i=1}^{N_v} P_i)$;

- $\forall\, P_i \geq 0\ \&\ \Sigma_{i=1}^{N_v} P_i > 1 : P_i = e^{P_i} / \Sigma_{i=1}^{N_v} e^{P_i}$.

### 4.5. Training Algorithm

The pseudoadversarial noise added to $X_c$ is generated in real time, resulting in differences in the training data of each epoch, thereby avoiding severe overfitting. In addition, a slight noise is beneficial for training effective detection models. This is one of the reasons why we artificially increase $P_0$.

To enable the classification model to distinguish adversarial examples (pseudoadversarial examples) from clean examples, the input images are replaced with JPEG errors. In specific

training steps, two additional strategies are designed to mitigate overfitting and enhance the detection performance. First, the amplitude of the pseudoadversarial noise is randomly reduced by half. We then design a quality factor fluctuation strategy. The compression and decompression quality factors used in the training examples are randomly reduced by one or two according to the test factor. The detailed training process is presented in Algorithm 3.

---

**Algorithm 3:** Process of training with pseudoadversarial noise

**Input:** $X_c$, $Noise_{ad}^p$, $Y_c$

1  **while** *epoch $\leq$ num_epoches* **do**
2  $\quad$ data0 = minibatch($X_c$) ;
3  $\quad$ noise = $Noise_{ad}^p$ / 2 * random.choice([1, 2]) ;
4  $\quad$ data1 = data0 + noise ;
5  $\quad$ **for** *i in range(len(data1))* **do**
6  $\quad\quad$ **if** *data1[i] > 255* **then**
7  $\quad\quad\quad$ data1[i] = 1 ;
8  $\quad\quad$ **end**
9  $\quad\quad$ **if** *data1[i] < 0* **then**
10 $\quad\quad\quad$ data1[i] = 0 ;
11 $\quad\quad$ **end**
12 $\quad$ **end**
13 $\quad$ factor = random.choice([$f + 2$, $f + 1$, $f_t$]) ;
14 $\quad$ data0 = data0 – $jpeg$(data0, factor) ;
15 $\quad$ data1 = data1 – $jpeg$(data1, factor) ;
16 $\quad$ data = cat(data0, data1) ;
17 $\quad$ label = cat(0, 1) ;
18 $\quad$ loss = $cross_e$($model$(data), label) ;
19 $\quad$ loss.$back$() ;
20 $\quad$ **if** *epoch > num_epoches $- N_t$* **then**
21 $\quad\quad$ loss = $cross_e$($model$($X_{ad}$), $Y_c$) ;
22 $\quad\quad$ loss.$back$() ;
23 $\quad$ **end**
24 $\quad$ epoch += 1 ;
25 **end**

---

It is worth noting that the JPEG compression and decompression processes is artificially added before being input to the model. Therefore, the quality factor is known, even during testing.

## 5. Results

### 5.1. Experimental Settings

5.1.1. Dataset Introduction

We used FGSM [10], BIM [23], Deepfool [18], C&W [12], DDN [19], BoundaryAttack [26], BrendelBethgeAttack [39], and BanditsAttack [25] to generate adversarial examples. For convenience, these attack methods are called FGSM, BIM, Deepfool, C&W, DDN, Boundary, Brendel, and Bandits, respectively. Among them, FGSM and BIM use coefficients of 2, 4, 6, and 8, and the Bandits method contains $L_2$ and $L_\infty$ versions, generating 15 subdatasets. The coefficient $\epsilon$ corresponds to the range from 0 to 255. Each subdataset contains 1000 examples to fit the noise distribution and 5000 for testing.

5.1.2. Experimental Environment

The experiments in this study are implemented using the PyTorch deep learning framework. The experimental hardware includes an Intel 12400 CPU and an NVIDIA RTX 4090 GPU. The software environment includes Ubuntu 22.04 LTS, Cuda 11.7, Python 3.10, and PyTorch 1.13.

*5.2. Performance Evaluation*

5.2.1. Performance of Different Models

Owing to structural differences across models, they perform differently in adversarial example detection. The performance of the model is not the focus of this study. Therefore, we designed a new indicator to evaluate the performance of the proposed dynamic simulation training strategy. The relative accuracy ($Acc_{rel}$) is defined as the ratio between the accuracy of the training strategy in this study and the accuracy trained with sufficient real examples of the same model, as shown in Equation (9). $TP_d$ and $TN_d$ represent the true positives and negatives of the proposed training strategy, respectively, and $TP_e$ and $TN_e$ are the true positives and true negatives of sufficient training examples, respectively.

$$Acc_{rel} = \frac{Acc_d}{Acc_e} = \frac{TP_d + TN_d}{TP_e + TN_e} \tag{9}$$

In some training detector schemes with real adversarial examples [13,17], the number of training examples is typically 50,000. Half of these are adversarial examples, and the other half are clean examples. Therefore, 50,000 training examples fit the above description for a sufficient number of real examples.

Based on the fast attack methods of FGSM and BIM, 25,000 adversarial examples were generated to build the training dataset. The performances of ResNet [40], WideResNet [41], DenseNet [42], VGG [38], and Res2Net [43] are listed in Tables 2 and 3.

**Table 2.** Performance of different models on the FGSM dataset.

| | | ResNet18 | ResNet34 | ResNet50 | ResNet101 | Res2Net50 |
|---|---|---|---|---|---|---|
| $\epsilon = 2$ | $Acc_d$ | 99.07% | 98.65% | 97.86% | 97.99% | 97.89% |
| | $Acc_{rel}$ | 100.39% | 99.87% | 99.85% | 100.21% | 99.65% |
| $\epsilon = 4$ | $Acc_d$ | 99.54% | 99.35% | 99.52% | 99.04% | 99.03% |
| | $Acc_{rel}$ | 99.83% | 99.63% | 99.88% | 102.61% | 101.23% |
| $\epsilon = 6$ | $Acc_d$ | 99.92% | 99.83% | 99.89% | 99.64% | 99.70% |
| | $Acc_{rel}$ | 100.03% | 100.01% | 100.05% | 99.81% | 100.41% |
| $\epsilon = 8$ | $Acc_d$ | 99.91% | 99.94% | 99.87% | 99.89% | 99.94% |
| | $Acc_{rel}$ | 99.98% | 100.02% | 99.98% | 99.96% | 100.02% |
| | | WideResNet50 | WideResNet101 | DenseNet169 | VGG11 | VGG16 |
| $\epsilon = 2$ | $Acc_d$ | 97.38% | 98.11% | 96.92% | 97.05% | 97.72% |
| | $Acc_{rel}$ | 98.47% | 99.03% | 99.28% | 98.22% | 99.45% |
| $\epsilon = 4$ | $Acc_d$ | 99.70% | 99.42% | 99.02% | 98.37% | 99.50% |
| | $Acc_{rel}$ | 99.83% | 99.56% | 99.61% | 98.75% | 100.13% |
| $\epsilon = 6$ | $Acc_d$ | 99.92% | 99.85% | 99.80% | 99.13% | 99.65% |
| | $Acc_{rel}$ | 100.20% | 99.96% | 99.98% | 99.31% | 99.80% |
| $\epsilon = 8$ | $Acc_d$ | 99.91% | 99.98% | 99.83% | 99.88% | 99.87% |
| | $Acc_{rel}$ | 100.02% | 100.01% | 99.90% | 100.23% | 100.03% |

**Table 3.** Performance of different models on the BIM dataset.

|  |  | ResNet18 | ResNet34 | ResNet50 | ResNet101 | Res2Net50 |
|---|---|---|---|---|---|---|
| $\epsilon = 2$ | $Acc_d$ | 96.25% | 96.69% | 95.78% | 96.13% | 95.84% |
|  | $Acc_{rel}$ | 98.96% | 99.21% | 98.55% | 98.65% | 98.67% |
| $\epsilon = 4$ | $Acc_d$ | 98.41% | 98.01% | 98.62% | 98.27% | 97.84% |
|  | $Acc_{rel}$ | 99.35% | 99.15% | 99.91% | 99.48% | 99.30% |
| $\epsilon = 6$ | $Acc_d$ | 99.28% | 98.96% | 99.08% | 98.81% | 99.06% |
|  | $Acc_{rel}$ | 99.85% | 99.59% | 99.99% | 99.63% | 99.74% |
| $\epsilon = 8$ | $Acc_d$ | 99.54% | 99.59% | 99.47% | 99.25% | 99.23% |
|  | $Acc_{rel}$ | 99.86% | 99.93% | 99.84% | 99.75% | 99.69% |
|  |  | WideResNet50 | WideResNet101 | DenseNet169 | VGG11 | VGG16 |
| $\epsilon = 2$ | $Acc_d$ | 96.47% | 96.09% | 92.16% | 95.96% | 92.40% |
|  | $Acc_{rel}$ | 98.90% | 98.51% | 100.45% | 98.66% | 121.77% |
| $\epsilon = 4$ | $Acc_d$ | 98.33% | 97.71% | 97.80% | 97.78% | 98.33% |
|  | $Acc_{rel}$ | 99.30% | 98.74% | 99.83% | 98.68% | 114.54% |
| $\epsilon = 6$ | $Acc_d$ | 99.21% | 99.45% | 99.46% | 99.30% | 99.07% |
|  | $Acc_{rel}$ | 99.54% | 99.73% | 100.65% | 99.84% | 101.29% |
| $\epsilon = 8$ | $Acc_d$ | 99.37% | 99.49% | 98.57% | 98.98% | 99.27% |
|  | $Acc_{rel}$ | 99.59% | 99.83% | 99.18% | 99.31% | 101.12% |

In Tables 2 and 3, for both the FGSM and BIM datasets, the $Acc_d$ of some models can exceed 96%. This indicates the superiority of the proposed dynamic simulation training strategy. The performance of each model differs significantly, even if the model structure differs only in the number of layers. Data from series models of ResNet suggest that too many layers are not beneficial.

With 1000 samples from each dataset, we calculated the influence of $\epsilon$ on the noise intensity, as shown in Table 4. The noise intensity ($Int_n$) is defined as the average noise amplitude of each pixel, as shown in Equation (10). $n$ represents the number of pixels in the picture. $X_{ad,i}$ and $X^c_{ad,i}$ represent the $i$th element of $X_{ad}$ and $X^c_{ad}$, respectively. For the same model, $Acc_d$ exhibits an upward trend with an increase in $\epsilon$. The change in $Acc_d$ is directly related to the noise amplitude: the greater the noise, the easier it is to detect the adversarial example. This trend is consistent with the results of existing adversarial example detection models.

**Table 4.** The effect of the perturbation coefficient ($\epsilon$) on the noise intensity ($Int_n$).

|  | $\epsilon = 2$ | $\epsilon = 4$ | $\epsilon = 6$ | $\epsilon = 8$ |
|---|---|---|---|---|
| FGSM | 1.99 | 3.97 | 5.93 | 7.91 |
| BIM | 1.38 | 2.29 | 3.15 | 3.68 |

$$Int_n = \frac{\sqrt{\Sigma_{i=1}^{n}(X_{ad,i} - X^c_{ad,i})^2}}{\sqrt{n}} \tag{10}$$

To analyze the training process of the detection model, the test accuracy of the BIM ($\epsilon = 2$) dataset in each epoch of ResNet34 training is counted as shown in Figure 7. The number of training epochs is set to 30, and $N_t$ is set to 10. This means that when training with the proposed method, only the last 10 epochs involve real adversarial examples.

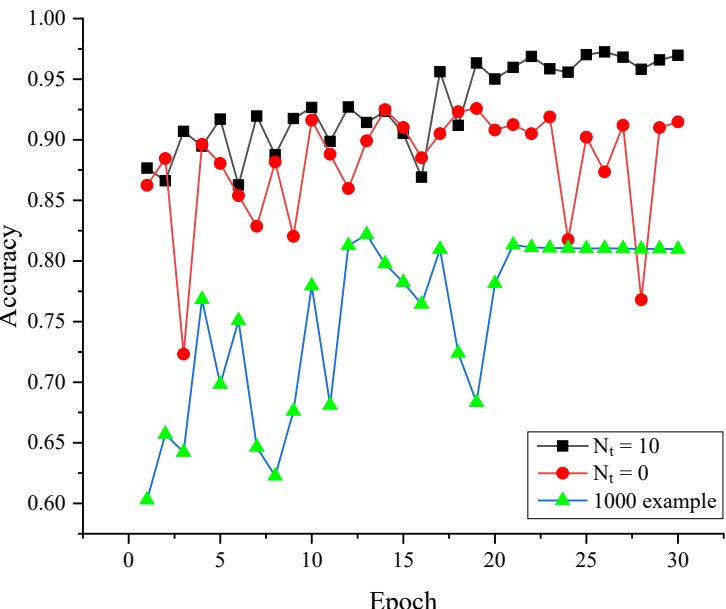

**Figure 7.** The test accuracy of ResNet34 in the training process with the whole proposed method, no $X_{ad}$, and $X_{ad}$ only.

Because the training dataset is randomly generated, the training accuracy of the model on it has no practical significance. Therefore, only the accuracy of the test dataset is shown in Figure 7.

The three curves in Figure 7 represent training using the whole proposed method ($N_t = 10$), no $X_{ad}$ ($N_t = 0$), and $X_{ad}$ only (1000 example), respectively. When only $X_{ad}$ training is used, the model suffers from serious overfitting, owing to the shortage of training data, which leads to limited accuracy. Compared with the data in Table 3, the accuracy of the 1000-example curve is significantly lower. In the $N_t = 0$ curve, the accuracy is much higher than that in the 1000-example curve. Owing to the high randomness of the training data, the $N_t = 0$ curve has a very large fluctuation, even in the last few cycles. With the help of $X_{ad}$, the $N_t = 10$ curve exhibits both high accuracy and small fluctuations.

### 5.2.2. Performance of Each Dataset

The detection performance of ResNet 34 with the dynamic simulation training strategy for each dataset is shown in Figure 8. Both the FGSM and BIM datasets contain only the case in which $\epsilon = 2$. To visualize the relationship between model detection performance and $Int_n$, Figure 8 also includes the $Int_n$ of each dataset. The lengths ($L_{Int}$ and $L_{Acc}$) of bins representing $Int_n$ and the accuracy were transformed using Equation (11) to make their relationship more obvious.

$$\begin{cases} L_{Int} = lg(1 + Int_n) \\ L_{Acc} = (Accuracy - 0.85)/0.15 \end{cases} \tag{11}$$

All accuracy in Figure 8 is more than 89%, indicating that the proposed dynamic simulation training strategy can help ResNet 34 achieve good results in the adversarial example detection task. The larger the $Int_n$ of the dataset, the better the performance of the detection model. This indicates that the detection difficulty of adversarial examples is positively correlated with the $Int_n$ of the examples generated by the current attack method.

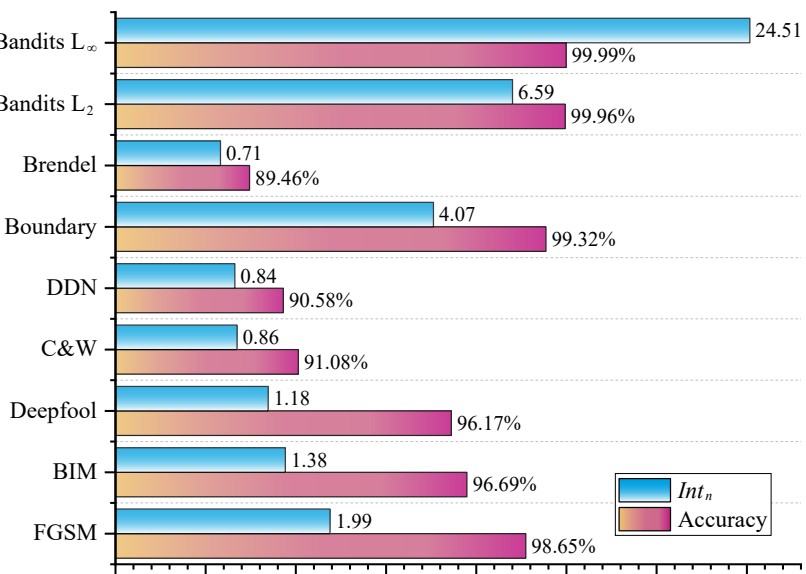

**Figure 8.** ResNet34 accuracy and $Int_n$ for each dataset.

### 5.3. Comparison and Analysis

To analyze the actual performance of the proposed training scheme, we compare it with ESRM [17], SmsNet [13], and DCT-Like [14] models with sufficient training data, as shown in Table 5. Since ESRM, SmsNet, and DCT-Like are not few-shot models, we trained them using 25,000 real adversarial examples. They cannot be trained with normal images alone. The model used in co-operation with the proposed scheme is ResNet34. For the remainder of this work, ResNet 34 is the default detection model in the proposed dynamic simulation training strategy.

**Table 5.** Comparison of the performance of different detectors.

|          | **Ours** | **ESRM [17]** | **SmsNet [13]** | **DCT-Like [14]** |
|----------|----------|---------------|-----------------|-------------------|
| FGSM     | 98.65%   | 98.10%        | 98.49%          | 99.64%            |
| BIM      | 96.69%   | 97.14%        | 99.27%          | 99.18%            |
| Deepfool | 96.17%   | 95.13%        | 98.26%          | 99.07%            |
| C&W      | 91.08%   | 92.87%        | 93.83%          | 95.02%            |

The proposed dynamic simulation training strategy does not require a large number of training samples. It achieve good results as a training scheme with only 1000 real adversarial examples. Owing to the differences between pseudo- and real adversarial examples and the limitations of the model, our solution still has a gap relative to other advanced detectors. However, the data in Table 5 indicate that this gap is within the range of 4%.

### 5.4. Experiments across Datasets

In addition to the model's performance on a single dataset, its performance on an unknown dataset is also an important metric. This means that we need to train with one dataset and test on different datasets. In this subsection, we train detection models with 1000 real adversarial examples, and the number of test images remains at 10,000.

#### 5.4.1. Cross-Coefficient Test

In Table 6, the FGSM and BIM datasets with perturbation coefficients ($\epsilon$) of 2 and 8, respectively, are used for training. While one $\epsilon$ is used for training, the others in 2, 4, 6, and 8 are used for testing. Table 6 shows the accuracy and recall rates of the tests.

**Table 6.** Test accuracy and recall across coefficients.

| Train \ Test | | $\epsilon = 2$ | $\epsilon = 4$ | $\epsilon = 6$ | $\epsilon = 8$ |
|---|---|---|---|---|---|
| | | | FGSM | | |
| $\epsilon = 2$ | accuracy | - | 99.07% | 98.78% | 98.92% |
| | recall | - | 99.91% | 99.82% | 99.86% |
| $\epsilon = 8$ | accuracy | 51.14% | 86.87% | 99.57% | - |
| | recall | 3.23% | 74.22% | 99.84% | - |
| | | | BIM | | |
| $\epsilon = 2$ | accuracy | - | 97.16% | 97.27% | 97.37% |
| | recall | - | 99.96% | 99.98% | 99.97% |
| $\epsilon = 8$ | accuracy | 85.89% | 92.61% | 98.99% | - |
| | recall | 83.32% | 86.59% | 99.65% | - |

When the training condition is $\epsilon = 8$, the detection accuracy of the $\epsilon = 2$ dataset is low. The accuracy of the FGSM dataset is approximately 50%, and the recall rate is only 3.23%. This shows that the cross-coefficient test effect of training with the $\epsilon = 8$ dataset is poor. When the training dataset is $\epsilon = 2$, the model detection performance is significantly improved, indicating that a small $\epsilon$ during training is efficient for unknown data detection.

### 5.4.2. Cross-Attack Method Test

In Figure 9, the performance of the proposed scheme tested across attack methods is shown as a heat map. The horizontal and vertical axes represent the training and testing datasets, respectively. Among the datasets, the $\epsilon$ of FGSM and BIM is 2, and Bandits is the $L_2$ version. The darker the color of the small square, the worse the test result.

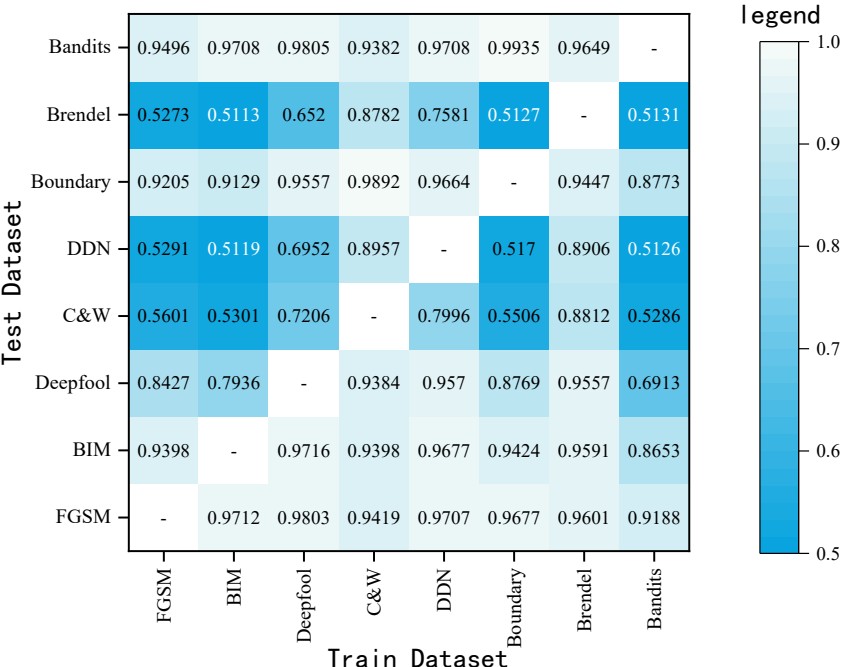

**Figure 9.** Test accuracy across attack methods. The "Legend" indicates the relationship between accuracy and colors.

The shade of the color for each column in Figure 9 indicates the cross-dataset test effect of training on one dataset. The lightest colored columns are the C&W and Brendel columns, which obtain the best detection model for these two datasets. In Figure 8, both C&W and

Brendel have low detection accuracy. The FGSM and Bandits that perform well, as shown in Figure 8, perform poorly in Figure 9. This shows that the cross-dataset performance is negatively correlated with the same dataset, and the detection effect of unknown data is negatively correlated with the $Int_n$ of the training dataset.

### 5.5. Multidataset Experiment

Although the cross-dataset detection capability can enable the detection of multiple types of adversarial examples using a single model (all parameters are the same), we still expect it to be able to detect multiple types of adversarial examples more efficiently. In this section, multiple $\epsilon$ datasets and multiple-attack method datasets are used to train and test the same model.

#### 5.5.1. Multicoefficient Test

The data in Table 6 show that the detection model is sensitive to changes in the perturbation factor ($\epsilon$). A decrease in the perturbation coefficient causes a rapid degradation in the model performance. Simultaneous training using multiple perturbation coefficients ($\epsilon$) is used to avoid this problem. The test accuracy of models trained with multiple $\epsilon$ values on the FGSM and BIM datasets are shown in Table 7, where overall represents a dataset consisting of all test samples with $\epsilon$ values equal to 2, 4, 6, and 8.

**Table 7.** Accuracy of the multicoefficient model. "Overall" represents a dataset consisting of all test samples.

|  | $\epsilon = 2$ | $\epsilon = 4$ | $\epsilon = 6$ | $\epsilon = 8$ | **Overall** |
|---|---|---|---|---|---|
| FGSM | 98.63% | 99.32% | 99.31% | 98.53% | 99.20% |
| BIM | 96.15% | 98.97% | 99.36% | 99.21% | 98.42% |

The accuracy in Table 7 is all greater than 96%. Compared with the single-dataset test performance in Tables 2 and 3, the gap of the multicoefficient model is within the range of 1%. This shows that the proposed dynamic simulation training strategy has good adaptability to multicoefficient datasets.

The multidataset model performs better than the single-dataset model on some datasets, such as the BIM dataset with $\epsilon = 4$ and $\epsilon = 6$. This is because more training data variations further avoid overfitting.

#### 5.5.2. Multimethod Test

The performance of the model trained on multiple attack-method datasets is presented in Table 8. Among the adopted datasets, the perturbation coefficient ($\epsilon$) of FGSM and BIM is 2, and Bandits is the $L_2$ version.

**Table 8.** Performance of the multimethod model.

|  | FGSM | BIM | Deepfool | C&W |
|---|---|---|---|---|
| Accuracy | 97.20% | 96.98% | 96.24% | 97.36% |
|  | DDN | Boundary | Brendel | Bandits |
| Accuracy | 87.82% | 96.81% | 84.08% | 97.12% |

The detection accuracy of the multimethod model exceeded 84%. Compared to the single-method model shown in Figure 8, the maximum gap is 5.38%. This gap is acceptable. However, its test effect on the Brendel dataset was 3.74% lower than that of the C&W dataset training model, as shown in Figure 9. Because multiple datasets work together to train a model, it tends to identify easy examples with a large $Int_n$ to quickly reduce

losses. This results in poor performance of the multimethod models on the DDN and Brendel datasets. However, the overall multimethod model detection performance is still satisfactory. The dynamic simulation training strategy effectively helps the model detect multiple adversarial examples.

## 6. Conclusions

In this study, we proposed a new task of training the adversarial example detection model when the data are insufficient and designed a dynamic simulation training strategy. In the proposed scheme, pseudoadversarial examples for training are generated in real time, which does not require additional time and extra costs. A small number of real adversarial examples is used to help the model achieve stable performance.

In the testing and analysis processes, we found that a small number of real adversarial examples played an important role. In future studies, more ways to use these models will be explored to further improve detection performance.

**Author Contributions:** Data curation, J.Z. and S.Q.; writing—original draft preparation, J.Z.; software, J.W. (Junfeng Wu); writing—review and editing, J.W. (Junfeng Wu), J.M.A. and J.W. (Jinwei Wang); visualization, S.Q. All authors have read and agreed to the published version of the manuscript.

**Funding:** This work was supported by the National Natural Science Foundation of China (62072250, U20B2065, 61872203, and 61802212), in part by the Plan for Scientific Talent of Henan Province (2018JR0018), in part by the Postgraduate Research & Practice Innovation Program of Jiangsu Province (KYCX200974), and the Priority Academic Program Development of Jiangsu Higher Education Institutions (PAPD) fund.

**Data Availability Statement:** Data sharing is not applicable to this article.

**Acknowledgments:** All authors would like to thank the editors and reviewers for their advice.

**Conflicts of Interest:** The authors declare no conflict of interest.

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
