# Peer review of "Detecting High-Resolution Adversarial Images with Few-Shot Deep Learning"

_remotesensing, doi:10.3390/rs15092379_

Round 1

Reviewer 1 Report

Adversarial attacks pose a significant threat to models based on deep learning. This work focuses on the problem of insufficient training adversarial images, which designs pseudo-adversarial examples generated through simulation. The reviewer still has some concerns.

1. The proposed training strategy used the JPEG compression method. Does it still work for other formats? For example, a png image?

2. Why should the noise values satisfy 0<Nv<511? 

3. In Section 5.5, how many training and test images are used in multi-coefficient test and multi-method test?

4. What does “distribution of $1$ in adversarial noise” mean? 

5. The author should give more explanations on the performance of the proposed method compared with DCT-like [12]. What is the advantage of the proposed method?

Reviewer 2 Report

In this paper, pseudo-examples are designed by capturing the distribution of adversarial noise and are used to train a detector with a small number of samples. The authors have taken the security of deep learning models in remote sensing images seriously and proposed a novel training strategy. Their work is innovative, and the experimental results demonstrate the superiority of the proposed scheme. Nevertheless, there are some minor issues in the paper that should be addressed, including:

1. In Section 4.1, the meaning of the symbol "~" should be interpreted. It means obey distribution, right?

2. In Fig. 4, The distance between the components is too large. A smaller distance might make the picture look better.

3. The meaning of the symbol "overall" in Table 7 should be explained in the table header. I saw the description in the text, but the header can help to make the table readable.

4. In Section 5.3, the comparison methods are not few-shot models. How did the authors set the parameters of the comparison models? For example, how many training samples were used?

The English is good, but there are some typos in the paper. The authors should check the English carefully.

Reviewer 3 Report

The authors considered the security of high-resolution image models and designed an adversarial image detection scheme for the case of insufficient samples. The proposed scheme of statistics-generation-training is novel. However, the writing of manuscript needs some modification.

1. In Section 2.1, authors mentioned "However, AdvGAN++ can only carry out white-box attacks because it requires internal features of the target model." What does "internal features of the target model" mean?

2. In the comparison and analysis part of experiments, the introduction of the comparison scheme and some important experimental parameters are lacking. The authors should further analyze the differences and advantages between the proposed and comparison scheme.

3. The symbol "Sober" in Eq. 7 and Eq. 8 is not introduced and it should be "Sobel."

4. Is the meaning of "ϵ" in the experiment the same as it in Eq. 2? Does it correspond to a normalized image or another value range?

5. The content of Fig. 9 should provide a more detailed introduction in the figure title, such as the purpose of "Legend".
